# Impact of Thermally Reducing Temperature on Graphene Oxide Thin Films and Microsupercapacitor Performance

**DOI:** 10.3390/nano12132211

**Published:** 2022-06-28

**Authors:** Vusani M. Maphiri, Daba T. Bakhoum, Samba Sarr, Ndeye F. Sylla, Gift Rutavi, Ncholu Manyala

**Affiliations:** Department of Physics, Institute of Applied Materials, SARChI Chair in Carbon Technology and Materials, University of Pretoria, Pretoria 0028, South Africa; vusanimuswamaphiri@gmail.com (V.M.M.); thiogna@yahoo.fr (D.T.B.); ssarr3112@gmail.com (S.S.); ntoufasylla@gmail.com (N.F.S.); rutavigift@yahoo.com (G.R.)

**Keywords:** graphene oxide, thermally reduced, Raman spectroscopy, sheet resistance, energy storage

## Abstract

In this work, a thermally reduced graphene oxide (TRGO) thin film on microscopic glass was prepared using spray coating and atmospheric pressure chemical vapour deposition. The structure of TRGO was analysed using X-ray diffraction (XRD) spectroscopy, scanning electron microscope (SEM), energy-dispersive X-ray spectroscopy (EDS), Fourier transform infrared (FTIR) spectroscopy, and ultraviolet–visible spectroscopy (UV-Vis) suggesting a decrease in oxygen functional groups (OFGs), leading to the restacking, change in colour, and transparency of the graphene sheets. Raman spectrum deconvolution detailed the film’s parameters, such as the crystallite size, degree of defect, degree of amorphousness, and type of defect. The electrochemical performance of the microsupercapacitor (µ-SC) showed a rectangular cyclic voltammetry shape, which was maintained at a high scan rate, revealing phenomenal electric double-layer capacitor (EDLC) behaviour. The power law and Trasatti’s analysis indicated that low-temperature TRGO µ-SC is dominated by diffusion-controlled behaviour, while higher temperature TRGO µ-SC is dominated by surface-controlled behaviour.

## 1. Introduction

Graphene has been widely studied and used in advanced technological applications due to its excellent properties, which make it a possible candidate for the fabrication of gas sensors [1], energy [2], and hydrogen storage devices [3], and transparent conducting electrodes for photovoltaic application [4]. Moreover, each application requires a different set of graphene properties. Thus, different graphene preparation methods are suitable for certain applications. For instance, while high-quality graphene (synthesised by chemical vapour deposition (CVD), bottom-up approach, low yield) is more suitable for electronic applications, it is not compatible with the production of conductive inks due to the lack of functional groups [5]. In addition, functionalised graphene prepared via the chemical exfoliation method (e.g., hummer’s method, top-down approach, high yield) of graphite is more suitable for energy-storage applications and other applications, such as gas sensors and polymer composites, because the oxygen functional groups (OFGs) on the graphitic surface act as polymers [6] or nanoparticle anchors [7], as well as increasing pseudocapacitive behaviour [2].

Graphene oxide (GO) was first reported by Brodie and later modified by Staudenmaier and Hummers [5]. These authors proposed that the chemical structure of GO consists of two main regions comprising hydrophobic conjugates C-sp^2^ and C-sp^3^ domains. Thus, GO contains alcohol and epoxy groups located at the basal plane, while carboxylic acids attaches at the surface edges [5]. The advantage of preparing graphene from the thermal reduction of GO is that physical, mechanical, electronic, and electrochemical properties can be easily modified by controlling the OFGs attached to the graphene sheets. Moreover, while thermal reduction does not leave residual compounds, the chemical reduction method using hydrazine hydrate has been reported to have residual molecules of N_2_H_4_ and incorporate NH_3_ within reduced GO sheets [8,9]. Chen et al. [10] and Zhao et al. [11] reported thermal reduction with GO and X-ray diffraction (XRD) spectroscopy, Fourier transform infrared (FTIR) spectroscopy, scanning electron microscopes (SEM), and energy-dispersive X-ray dispersion spectroscopy (EDS), which were used to analyse the structure and surface morphology. It was concluded that the XRD showed a decrease in the interplanar distance, while the FTIR and EDS showed a decrease in the oxygen functional groups. The SEM showed the presence of a sheet-like material, which was similar for pre- and post-reduction. The electrochemical performance showed an increase in the power density and a decrease in the energy density as a function of RT [11]. This behaviour was allotted to the removal of resistive OFGs, i.e., the carboxylic acids attached to the surface edges resisting the flow of electrons [5] (confirmed by the increase in electrical conductivity as the reducing temperature increased). These OFGs also served as a passage for the ions to the bulk internal surface, as well as facilitating the fast redox processes occurring at or near the electrode surface, which can provide pseudocapacitance, as shown by our previously reported results [2,12]. Our electrochemical impedance spectroscopy (EIS) results showed a decrease in the solution resistance, further confirming the removal of the surface-edges functional group. 

The shape of the electrochemical cyclic voltammetry (CV) has been used to distinguish between different electrochemical systems, i.e., capacitive and faradic behaviour. Further electrochemical analysis, such as Trasatti’s [13,14] and Dunn’s [14] analysis, can be used to understand the electrode material storage mechanism. Trasatti’s analysis relies on the assumption that diffusion and surface-controlled contributions are controlled by different kinetics, which respond differently as the scan rate increases. Xu et al. [14] used this analysis on the prepared nitrogen-rich holey graphene oxide film reduced at 200–400 °C under an Ar environment. The Trasatti analysis showed that for an optimum sample reduced at 300 °C, the capacitance contribution from the inner surface (pseudocapacitance) is almost 64.1% of the total capacitance. Huang et al. [13] used Trasatti’s method to understand the different CV curve calculated capacitance levels generated from the three prepared electrodes, i.e., c-TiO_2_ + MWNT, p-TiO_2_ + MWNT, and p-TiO_2_ + c-TiO_2_ + MWNT. The results showed that the presence of c-TiO_2_ resulted in a decrease in the total capacitance due to pore blockage (c-TiO_2_ = 20 nm). 

This study is a continuation of our previously published work on TRGO thin film on microscopic glass (MSG) for microsupercapacitor applications [12]. Here, we focus mainly on the structural, electrical, optical characterisation, and electrochemical analyses, such as the electrolyte-ion diffusion of the TRGO µ-SC. In addition, this study presents the fundamental understanding of the influence of oxygen on the surface of graphene sheets and its impact on the electrochemical performance. Similar studies have been reported on powder GO [10], in which the GO exfoliated at around 200 °C, while the GO within this study changed colour from yellowish to dark black at 200 °C and then turned to shiny black as the temperature increased. This work offers a pioneering characterisation of TRGO thin films prepared in this fashion, i.e., an inexpressive, simple, and quick way to fabricate TRGO thin films that does not contain chemical residue from chemical reduction. These thin films can be used in various applications, such as energy storage [12], counter electrodes for dye-sensitised solar cells [15], and temperature sensors [16]. 

## 2. Experimental Methods

### 2.1. Materials

Chemicals used in this article were of analytical grade and were used without any further purification. The materials and chemicals used were as follows: natural graphite, sulphuric acid (H_2_SO_4_ (98%)) (Associated Chemical Enterprises, Johannesburg, South Africa), microscopic glass (MSG) (B&C Glass Ltd, Haverhill, UK), potassium permanganate (KMnO_4_) %)) (Associated Chemical Enterprises, Johannesburg, South Africa), ethanol (C_2_H_5_OH (98%)) %)) (Associated Chemical Enterprises, Johannesburg, South Africa), hydrogen peroxide (H_2_O_2_ (50%)) %)) (Associated Chemical Enterprises, Johannesburg, South Africa), Argon gas (Ar (99%)) (AFROX, Johannesburg, South Africa) and deionised water (H_2_O) (DW) (prepared using DRAWELL, laboratory water purification system, available at the Physics Department, University of Pretoria, Pretoria, South Africa) at 18.2 MΩ.

### 2.2. Preparation of TRGO Thin Film 

TRGO thin films were prepared using a series of methods that were modified Hummer’s method, cleaning MSG using piranha solution, spray-coating GO on the cleaned MSG, and then thermally reducing GO using atmospheric pressure chemical vapour deposition (AP-CVD) as per our previously published results [12]. In summary, graphite powder was added into an agitating cooled H_2_SO_4_, followed by KMnO_4_. The agitation continued for 180 min at a constant heating temperature of 50 °C. The solution was left to cool to room temperature, then DW and H_2_O_2_ were added into the solution. GO was cleaned via decantation, then centrifuged at 5000 rpm for 5 min, and dried at 60 °C for 3 h. Dried GO was mixed with ethanol, then sonicated to obtain a GO-ethanol which was sprayed onto a clean MSG. The GO/MSG thin film was subsequently reduced at a temperature ranging from 100 to 500 °C. Note that the samples were denoted as TRGO-100, TRGO-200, TRGO-300, TRGO-400, and TRGO-500, depending on the reducing temperature used. These preparation methods and appearance of the prepared thin film are schematically illustrated in Figure 1. The digital images of the prepared thin film are also displayed in Appendix A.

### 2.3. Structural, Morphological, and Electrochemical Characterisation

The prepared TRGO thin-film samples, together with precursor graphite and intermitted GO and MSG substrates were analysed using various techniques: Bruker BV 2D phaser benchtop X-ray diffraction (XRD) (PANalytical BV, Amsterdam, The Netherland) using CuKα radiation source was used to investigate the crystal structure. Zeiss Ultra-plus 55 field emission scanning electron microscope (FE–SEM) (Akishima-shi, Japan) operated at voltage of 2.0 kV was used to capture the surface morphology (SEM micrograph) and Oxford energy-dispersive X-ray spectroscopy (EDS) operated at 20.0 kV and controlled using Aztec 3.0 SP1 software was to obtained elemental information. Confocal WITec alpha 300RAS+ (Ulm, Germany) Raman microscopy was used to obtain the Raman spectra and the atomic force microscope (AFM) images. The Raman spectra were collected using a 532-nanometer excitation laser (the laser power was set to less than 2 mW to avoid heating the sample) at a single accumulation for 60 s at room temperature. The AFM images were collected using the tapping mode configuration and a silicon tip on a 30 × 30 μm^2^ area to examine surface topography of the GO and TRGO at different reducing temperatures. The three-dimensional AFM images and height profile were prepared using the WITec project five (build 5.1.18.79). Fourier transform infrared (FT-IR) spectroscopy was performed using the Bruker Alpha platinum-ATR (attenuated total reflection) (Billerica, MA, USA) in the range of 4000 to 400 cm^−1^. The light transparency and absorption effect of the prepared samples were examined by the transmittance and absorption spectra in wavelength range of 200–1000 nm using the Agilent cary 60 UV-Vis spectrophotometer (Santa Clara, CA, USA). The transmittance and absorbance of the substrate glass was taken first after the completion of the baseline correction with air background. The sheet resistance of the TRGO films was measured using the Ossila (Sheffield, UK) four-point probe (4PP) set-up, with a collinear and equidistant geometrical configuration. The two-electrode configuration measurements of the µ-SC device were measured using cyclic voltammeter (CV), galvanostatic charge-discharge (GCD), and electrochemical impendence spectroscopy (EIS) at 10 mV in a frequency range of 0.1 to 100 kHz using the Bio–Logic VMP-300 potentiostat (Knoxville, TN, USA) monitored by EC-Lab V11.33 software (Edmonton, AB, Canada).

## 3. Results and Discussion

### 3.1. X-ray Diffraction Spectroscopy

The XRD was deployed to investigate the structural properties of the thermally reduced samples, together with the precursor graphite powder, intermediate GO thin film, and MSG substrate. The obtained XRD patterns are displayed in Figure 2 and Appendix A. The pattern were indexed to the XRD patterns of GO (JCPDS 41 1487) and graphite (JCPDS 75 2078). The XRD diffraction peak of the graphite at 2θ≈ 25° resulted from the (002) reflection plane, which completely disappeared upon oxidation and evolved into a new peak at a diffraction angle of 2θ≈15°, which was indexed to the (001) reflection of the GO [17,18]. This indicated the successful oxidation of graphite, which was in line with the results reported in several other studies [11,19]. The absence of additional peaks on the GO, except the peak indicated with a Daggar (†), suggests an impurity-free GO. The diffraction peak assigned to † was the diffraction from the microscope glass substrate (MGS; see Appendix A, comparing the diffraction patterns of the GO and the MGS). When the TR temperature increased, the XRD pattern peaks changed back from the (001) to (002) reflection planes. The (002) peak shifted from around 2θ = 15 (TRGO-100) to 25° (TRGO-500), which suggests that the GO restacked (decreasing the interplanar distance) into a disordered graphite-like material due to the removal of the OFG, increasing the van der Waals attraction force [11,18]. This behaviour can be seen on the SEM and AFM images displayed in Appendix A and the roughness information is displayed in Appendix A. In addition, similar results were reported by Zhao et al. [11], in whose study the diffraction peak also shifted as a function of the RT. The diffraction peaks assigned to the asterisk (⁎) were due to the diffraction peaks of both the substrate and the GO/graphite-like material. These diffraction patterns are clearly shown in Appendix A, where the diffraction pattern of the TRGO on the MGS were compared to the annealed MGS. We suggest that the diffraction peak denoted with a bullet (•) represents the highly reduced GO at the MSG interface caused by the prolonged heating exposure from the hot-glass substrate during cooling. This peak cannot be attributed to the MGS-related diffraction peak because the diffraction pattern of the MSG depicted in Appendix A does not resemble such a peak. On the TRGO-500 (see also Appendix A), the • denoted peak is not visible due to the homogenous reduction caused by the high reduction temperature. The interplanar distances of the (001) and (002) diffraction peaks were estimated using the Bragg’s Equation (1) and depicted in Appendix A.
(1)nλ=2dsinθ
where n, λ, d, and θ are the integers, the wavelength of the Cuk_*α*_, the interplanar distance at (001) and (002) miller indices, and the diffraction angle in rad, respectively. The above-mentioned dramatic shift to a lower diffraction angle can also be explained in terms of the interplanar distance caused by the insertion of H_2_O molecules within the layers and the attachment of O-containing functional groups, such as hydroxyl (-OH), carbonyl (-CO-), carboxylic (-COOH), and epoxy (-O-) on the graphene layers within the graphite [11,20]. Similar results were obtained in the study by Bao et al. [21], where GO was also synthesised by the improved Hummer’s method and the obtained GO had an interspacing distance of around 0.803 nm. As a result of the RT ranging from 100–500 °C, the diffraction angle shift to a higher diffraction angle corresponded to the contraction of the interplanar distance allotted to the thermal removal of the H_2_O and O-containing functional groups. These results are confirmed by the FTIR displayed in Appendix A, showing the presence of H_2_O- and oxygen-related functional groups on the GO. The FTIR spectrum notches related to the H_2_O and OFGs fade as the temperature increases, which further confirms the XRD behaviour.

### 3.2. Energy Dispersive X-ray Spectroscopy

The elemental composition of the graphite, GO and TRGO-100 to TRGO-500 was obtained via the EDS. The obtained EDS spectra of these samples are presented in Figure 3. The spectra show the presence of multiple peaks attributed to various elements and/or the same element with different characteristic X-rays [22]. The graphite sample is indicated by the dominant peak at 0.277 KeV, which is due to carbon. This is because graphite is predominantly a carbon-based material. In order to determine the origin of the unexpected peaks within the spectra of the GO and TRGO samples, the EDS of the MSG were also measured, and they are displayed in Figure 3. Most of the peaks seen in the GO and TRGO spectra represent the MSG. The peaks at 0.525, 1.041, 1.254, 1.487, 1.740, and 3.692 KeV are due to the $k_{\alpha }$ characterisation peaks of oxygen (O), sodium (Na), magnesium (Mg), aluminium (Al), silicon (Si), and calcium (Ca), respectively. The EDS spectra of the GO and TRGO possess two peaks in addition to those of the MSG due to carbon (C) and sulphur (S) occurring at 2.308 KeV. Even though EDS is not a quantification technique, it is interesting to see that as the RT increases, the carbon and oxygen peaks increase and decrease, respectively, further confirming the XRD and FTIR results. Appendix A indicate the carbon content measured by the increase in the intensity of the carbon over that of the oxygen as a result of the extermination of the OFGs.

### 3.3. Optical Studies

The optical studies were measured with the aid of the UV-Vis. The absorbance and transmittance of the MSG are displayed in Appendix A. It can be seen that the MSG is highly transmissive in the visible and infrared (IR) region and only absorbs in the ultraviolet (UV) region [23]. The absorbance and transmittance of the GO and TRGO samples in Figure 4 are presented from the visible to the IR region. The GO and TRGO-100 absorbs (Figure 4a) highly in the visible range, which is confirmed by the zero transmittance within the same region. The hump around 300 nm can be attributed to the $n\;\to \;\pi^{*}$ transitions of the carbonyl (C=O) groups on the GO and TRGO-100 [24,25,26]. The lack of a hump at 300 nm after the 200 °C RT shows a successful oxygen reduction. Thus, the TRGO samples were reduced at 200 °C. Unlike the reduced GO oxide dispersed in water, which shows a similar absorption curve to that of the GO, albeit at a slightly increased absorption intensity (see absorption UV-Vis in [24,25]). The TRGO with RT ≥ 200 °C has a linear absorption curve similar to that of graphite [26]. Figure 4b shows that the GO and TRGO-100 have transmittance in the infrared region, unlike the other TRGO samples. These results confirm the appearance of the prepared thin films displayed in Appendix A, where the GO thin film is transparent and yellow-brownish, while the TRGO-100 is also semi-transparent and brown-blackish [12,19]. The other TRGO samples prepared at above 200 °C are opaque and dull black turning to shiny graphite grey. This further confirms that the reduction process occurred at 200 °C. 

### 3.4. Raman Spectroscopy

Raman spectroscopy was used to analyse the structural changes that occurred during the oxidation from graphite to GO and the reduction from GO to TRGO. The spectra of the natural graphite powder and GO/TRGO were measured at an excitation wavelength of 532 nm at room temperature and are displayed fully in Appendix A, while the one-phonon peaks are displayed in Figure 5a. In general, the obtained spectra exhibited peaks occurring in both one- and two-phonon Raman regions, D, G, 2D, D+G, and 2D′, occurring at around 1363, 1604, 2714, 2936, and 3197 cm^−1^, respectively. The D peak is associated with the local defects and disorder of the edges of the graphene and graphite materials, while the G peak is due to the highly ordered hexagonal structure within the graphite [17,19]. The 2D is an overtone of the D peaks, which is a fingerprint of the graphene-sheet formation and number of layers. The D+G and 2D′ are the overtone peaks of the combination of the D, G, and D′ peaks [27]. Upon the oxidation of the graphite, the Raman spectra of the GO showed significant differences bordering the one-phonon-range peaks. The broadening of the D peak from the full width at half maximum (FWHM) of 34.9 to 117.1 cm^−1^ is due to the defects caused by the attachment of the O-containing functional groups on the graphitic sheets in the oxidation reaction process. The increase in the FWHM of the G peak from 20.1 to 66.9 cm^−1^ is also due to the bond-angle disorder caused by the attachment of the O-containing functional groups. This caused the average ideal graphite-like hexagonal 120° bond angle to change [28] (see Figure 6a). As the temperature increases, there is no significant difference in the overall spectra, except the fading of the overtone peaks, which was probably due to restacking, as seen on the XRD, and the damage of the graphene sheets due to the defects. The FWHM of the D and G peaks seemed not to decrease, regardless of the removal of the OFGs. This was also observed by Claramunt et al. [29] when reducing the GO in the temperature range of 100 to 900 °C. The fading of the overtone peaks suggests that the layers increased (or restacked), since they only appear from graphitic crystallites containing few layers of graphene sheets. Thus, the stacking increased as temperature increased. As seen on the XRD, the interspace distance decreased, implying a high level of stacking, which will have led to low 2D intensity. Papani et al. [30] showed that the quality of graphene sheets is determined by the presence of 2D peaks; thus, the absence of a 2D peak suggests highly defective graphene sheets similar to the Raman spectra of activated carbon [31,32].

The Raman spectra in Figure 6a show that there is no D-peak shift, since it has been shown that Raman is insensitive to oxygen-related defects, such as -OH and -OOH [33]. The G band has a slightly red shift, which was attributed to the oxygen reduction, since this peak shifted to the oxygen-free graphite peak [29,34]. This was confirmed by the XRD and EDS, since they also showed the increasing removal of the OFGs as a function of the increasing RT. The Raman spectra of these samples were deconvoluted using the Lorentzian function in the D and G regions and are displayed in Figure 5b–h. The obtained additional peaks, labelled D*, D, and D′ were attributed to the lattice vibration of the sp^2^-sp^3^ bonds and phonon modes in the density of states of the graphitic crystallites, the amorphous carbon in the interstitial sites of the carbon lattice, and the carbon-lattice vibrations corresponding to that of the G band, respectively [28,35]. Note that the D*, D″, and D′ peaks were not observed in the graphite spectrum. The ratios (measured by the decrease in the X/G, where X = D, D*, D″, D′, and D/D′) of the area under the different deconvoluted peaks are displayed in Figure 6b–d. The D/G displayed in Figure 6b shows an increase from the graphite to the GO because of the defects during the reduction process. Upon the reduction at 100 °C, the graph increased because of the vacancy caused by the removal of the intercalated H_2_O, the low thermal OFGs, and the crumpling of the graphene sheets. The remaining functional groups also contributed to the D band. Figure 6b shows the crystallite size (L_a_) determined by the Knight formula using Equation (2) [28,31]
(2)$$L_{a}=\;C\left(\lambda \right)/\left(D/G\right)$$
where $C\left(\lambda \right)$ is the wavelength-dependent pre-factor determined from the $C\left(\lambda \right)\approx {\;C}_{0}+{\lambda C}_{1}$ for the wavelength in the visible range (400 nm< λ < 700 nm) $C\left(\lambda \right)=4.96$ nm for λ=532 nm, $C_{0}$ and $C_{1}$, which were estimated to be around −12.6 nm and 0.033, respectively [28,31]. The crystallite size decreased dramatically from 22 (graphite) to ~1.5 nm (GO/TRGO-100) due to the graphene-sheet breakdown during oxidation via Hummer’s method. The crystallite size (sp^2^ cluster) also increased as the RT increased, suggesting a slight structural recovery. The D″/G and D′/G are displayed in Figure 6c. The amorphous carbon (D″/G) increases from the GO to the TRGO-100 were due to the removal of OFGs, which resulted in damage to the graphene sheets. When the RT increased beyond 100 °C, the degree of amorphous carbon reduced, further confirming the structural recovery of the graphene. The ratio of the defect-activated mode (D′) relative to the graphitic carbon (G) (D′/G) shows an increase between the GO and the TRGO-100 (i.e., from 0.24 to 2.07) and a decrease to 0.36 in the TRGO-500, showing that the amount of defects reduced. Moreover, studies have shown that defect-activated modes (D/D′) (Figure 6d) are proportional to the defect concentration (*n_d_*). The *n*_d_ can be estimated from the assumption that I_D_~A_d_*n*_d_ and I_D′_~B_d_*n*_d_, where A_d_ and B_d_ are constants, depending on the type of perturbation introduced by the defect in the crystal lattice (or the nature of the defects). Thus, I_D′_/I_D_~A_d_/B_d_, and the ratio can be used to acquire the information on the nature (or types) of defects. If I_D′_/I_D_ ≅ 13, the defects are associated with sp^3^ hybridisation for vacancy-related defects I_D′_/I_D_ ≅ 7 and ≅ 3.5 for boundary defects [28,36]. Thus, the TRGO samples featured various defects.

### 3.5. Electric Studies

The electrical properties of the TRGO thin film at various temperatures were measured using the four-point-probe (4PP) method [37,38]. The sheet resistance ($R_{s}$) measurement was determined using Equation (3)
(3)$$R_{S}=4.53\times \frac{V}{I}\;(\Omega /\mathrm{sq}.)$$
where 4.53 is the thickness-correction constant, V is the voltage in volts, and I is the current in amperes between probes, respectively. The sheet resistance as a function of RT is displayed in Figure 7a. Note that the sheet resistance for GO and TRGO-100 could not be determined due to the high oxygen content and intercalated water. This is in line with the FTIR and EDS displayed in Appendix A and Figure 3, which show that the GO and TRGO-100 had pronounced notch and oxygen peaks due to high concentration of OFGs compared to those reduced at 200 °C and above. These results are also confirmed by the UV-Vis in Figure 4, showing the presence of the carbonyl (C=O) groups on the GO and TRGO-100. The UV-Vis shows the correlation between the transparent thin films; the light green area is transparent and the dark green is opaque at 550 nm. It can be seen that the sheet resistance decreases with the increase in RT. This is due to the removal of the surface-attached OFGs, such as -OH. Thus, the TRGO-500 has the lowest oxygen content, which causes the least resistance. Figure 7b shows the possible relationship between the carbon content in the EDS and the amorphous (D″/G) of the TRGO-200 to TRGO-500 for the sheet resistance. It can be easily inferred that a higher degree of lower carbon content, i.e., higher oxygen content (C/O), increases the resistivity. Such results have been also observed in non-carbon-based materials. Marciel et al. [39] showed that MoO_x_ thin film grown at a higher O_2_/Ar flow rate is more resistive compared to that grown in the absence of O_2_. It is clear that the carbon content is directly proportional to the sheet resistance and indirectly proportional to the degree of amorphousity. This behaviour can be also attributed to the slight structural recovery shown in Figure 6, according to which the sp^2^ cluster (L_a_), degree of defect (D/G), and amount of defects (D′/G) decreased to those of the graphite.

### 3.6. Electrochemistry

The electrochemical performance of the prepared TRGO thin films was tested on a microsupercapacitor (µ-SC) configuration, displayed in Appendix A. The µ-SC schematic is displayed in Figure 8a. The detail of the schematic is presented in Appendix A. Note that the detailed preparation and fabrication µ-SC is shown in our previous study [12], which also elucidated that the optimum capacitance was obtained at 14 digits per unit area due to the balance of the following factors: the average ionic diffusion pathway between adjacent digits [40,41]; the distributed capacitance effect, which suggests that higher digits lead to superior electrochemical performance [42]; the removal of the active electrode mass, which increases as the number of digits increase, resulting in reduced capacitance and an increase in the electric field strength around the edges of the interdigitated electrode. It is well known that the electric field line strength increases for edge-intensive electrodes rather than continuous or round-shaped electrodes [43,44]. The measured (CV) curves of the prepared µ-SC from the TRGO samples are displayed in Appendix A. The rectangular- shaped CV curves clearly indicate electrochemical double-layer capacitance (EDLC) characteristics within a working potential of 0.8 V. The CV curves show an increase in the scan rate relative to the RT (Figure 8b). This behaviour could be attributed to the increase in conductivity caused by the removal of the OFGs. Figure 8c illustrates the areal capacitance calculated from the CV curves in Appendix A at various RT using Appendix A, and the calculated value are also displayed in Appendix A. It can be clearly seen that the areal capacitance decreased as the RT increased. This can also be attributed to the decrease in the oxygen content, which has been shown to enhance the pseudocapacitive behaviour of graphene oxide [45], provide high wettability [19], and serve as a passage for ions into the inner bulk material [11]. The high interplanar distance determined from the XRD also provides easy access for the ions within the µ-SC material. The areal capacitance also dramatically decreases for µ-SC with higher oxygen content compared to those with lower oxygen content due to a lack of interaction time. This suggests that high-oxygen-content µ-SC makes a higher contribution of diffusion-controlled behaviour. Thus, oxygen-deprived TRGO has a better rate capability.

To extensively analyse the contributions of surface- and diffusion-controlled behaviours within the prepared TRGO µ-SC, the CV curves displayed in Appendix A were utilised to obtain the values of the current (i) corresponding to the scan rate (v). According to the power law ($i=av^{b}$), a and b are adjustable values, whereas the b value of 0.5 indicates that the current is controlled by diffusion-controlled behaviour, and the b value of 1 indicates that the current is controlled by surface-controlled behaviour. The graph of the log scan rate versus the log current is displayed in Appendix A, while Figure 8d illustrates the b values versus RT. These figures signify an increase in the dominance of the capacitive-controlled process as the RT increases. This is due to the decrease in OFGs limiting the diffusion-controlled behaviour [46]. 

A Trasatti analysis (Appendix A) was performed to calculate the pseudocapacitance and EDL capacitance contributions. The maximum total capacitance (C_T_) was obtained from the reciprocal of the extrapolated intercept of the 1/C_T_ versus v^0.5^ and the maximum EDL capacitance (C_EDL_) was estimated from the extrapolated intercept of C_EDL_ versus v^−0.5^, while the maximum pseudocapacitance (C_p_) was obtained from the difference between C_T_ and C_EDL_ (Appendix A). The obtained values are displayed in Appendix A and Figure 8e. It can also be seen that the capacitive mechanism dominated as the RT increased. The GCD curves are displayed in Appendix A; they show an isosceles-triangle-like curves, indicating EDLC behaviour. Thus, these results coincide with the CV curves and their analysis. The areal capacitances were also calculated from the GCD, and the obtained values are displayed in Appendix A. Appendix A shows similar behaviour to that of the areal capacitance from the CV curves versus the RT, displayed in Figure 8c. The areal capacitance also decreased, as mentioned above, and, interestingly, the high-temperature-reduced µ-SC can function at an extremely low current density compared to the low-temperature-reduced µ-SC, due to its low electrical resistance, as determined by the 4PP. The GCD stability of the four TRGO samples was analysed, and the results are displayed in Appendix A. The capacitance columbic efficiency was 100% throughout the measurement, while the capacitance retention, also displayed in Figure 8f, showed an increase as the RT increased due to the removal of the oxygen during the electrochemical stability measurement [47]. These results are in agreement with the studies reported by Yang et al. [48,49], which showed the possibility of electrochemically reducing graphene oxide by repetitive cathodic cycling, which resulted in the elimination of electrochemically unstable functional groups. Thus, the high capacitance retention can be attributed to the maintenance of the structural integrity of the µ-SC active material after continuous GCD cycling. The areal energy ($E_{areal}$) and power ($P_{areal}$) densities were calculated using Appendix A, respectively. The TRGO-200-to-TRGO-500 µ-SCs delivered areal power ranging from 0.3316 to 0.3709 mW cm^−2^, corresponding to areal energy in the range of 0.1368–0.0017 mW h cm^−2^ at various areal current densities. It is very clear that the delivered areal energy decreased as the reducing temperature increased, while the power increased due the removal of pseudocapacitive OFGs and increased conductivity. Note that the electrochemical performances of the prepared µ-SC are displayed on Table 1 and also compared with the performances of other µ-devices. 

Furthermore, the TRGO µ-SCs were also analysed using the EIS. The obtained Nyquist plot is displayed in Figure 9a, while the high-frequency zoom of the Nyquist plot is displayed in Appendix A. The Nyquist plot indicates that the TRGO-200 to TRGO-500 had a solution resistance (R_s_) of 4705, 3595, 2584, and 2326 Ω, respectively. The R_s_ decreased as the RT increased due to the removal of the resistive OFGs on the edge of the GO sheets. The R_s_ was used to calculate the maximum specific power ($P_{max}$) using Equation (4). The maximum specific energy ($E_{max}$) was calculated using Equation (5).
(4)$$P_{max}=\frac{V^{2}}{4\times \left(ESR\right)\times A_{area}}$$
(5)$$E_{max}=\frac{C_{T}V^{2}}{7.2\times A_{area}}$$
where V, ESR, $A_{area}$, and $C_{T}$ are the operating potential (v), equivalent series resistance (Ω), total active µ-SC area (cm^2^), and total capacitance (F cm^−2^) from the Trasatti analysis displayed in Appendix A. The obtained values for the maximum specific power and maximum specific energy are displayed in Appendix A. 

The Nyquist plot shows that the shape of the TRGO sample is similar to that of the porous carbon supercapacitors [54]. The plot does not show any semi-circular regions, indicating that the TRGO materials possessed very low charge-transfer resistance [54,55]. The plot also intersects with the real axis at 45°; this was attributed to Warburg impendence, in which electrolyte ions diffuse into porous electrodes, characterised by the linear part of the sloped line parallel to y-axis. A pure capacitor should exhibit a vertical line at low frequency, and the deviation from the vertical line is attributable to the diffusion resistance of the electrolyte ions [56]. The Bode plot displayed in Figure 9b shows the dependence of the impedance phase angle on the frequency of the TRGO samples. The phase angle increased as the temperature increased from −48 to −85°, which is close to the ideal value of the full capacitive behaviour of any supercapacitor [47,54]. The frequency dependence of the real and imaginary part of the capacitance (Re C and –Im Z) were evaluated using the complex capacitance model presented in Appendix A [57,58] and displayed in Appendix A. The Re C signifies the real accessible capacitance of the µ-SC and Im C corresponds to the energy loss by the irreversible process. The Im C in Figure 9c shows the peak frequency (knee frequency), which corresponds to the relaxation time ($\tau_{o}=f_{0}^{-1}$), indicating the minimum time required to charge/discharge the µ-SC with an efficiency greater than 50% [47,58]. The TRGO-500 µ-SC showed the lowest relaxation time compared to the other TRGO µ-SC. In Figure 9a, the Nyquist plot shows a slightly inclined line, which is attributable to the diffusion of the electrolyte ions into the bulk of the active electrode material, also referred to as Warburg diffusion. The Warburg coefficient $\sigma_{w}$ was estimated using the Equation (6) [59]:(6)$$Re\;Z=R_{e}+R_{ct}+\sigma_{w}\omega^{-0.5}$$
where $R_{e}$, $R_{ct}$, Re Z, and $\omega^{-0.5}$ are the electrolyte resistance, charge-transfer resistance, real impendence corresponding to the angular frequency in the low-frequency region and the square root of the lower angular frequency. Note that the $R_{e}$ and $R_{ct}$ parameters are independent of the frequency. Thus, the slope obtained from the Re Z dependence on the reciprocal square root of the lower angular frequencies ($\omega^{-0.5}$) correspond to the Warburg coefficient $\sigma_{w}$. The obtained values are displayed in Appendix A and Figure 9d, and replotted for clarity in Appendix A. It can be observed that the TRGO-200 had the lowest Warburg coefficient $\sigma_{w}$ of 12.4831 Ω cm^2^ s^−0.5^, compared to the other µ-SCs. Thus, the TRGO-200 had the highest ion diffusion coefficient of 4.3075 × 10^−13^ cm^2^ s^−1^, since the Warburg coefficient is directly proportional to the ion diffusion coefficient (Appendix A). The TRGO-200 had a higher ion diffusion because many OFGs were present on the surface of the TRGO-200 graphene sheets, i.e., hydroxyl, carboxyl, and epoxy groups, which served as passages to the internal surface for the ions [11]. In addition, the XRD showed a large interplanar distance on the TRGO-200. This allowed the easy diffusion of ions into the internal surface.

## 4. Conclusions

A detailed investigation of the GO/TRGO thin film on MSG following different reduction temperatures was presented. This study follows from our previous work, which mainly focused on microsupercapacitor applications [12]. It was demonstrated that the XRD-calculated interspace distance decreased, while the EDS also showed an increase in carbon content as the RT increased. The amorphous carbon (I_D″_/I_G_) and degree of defect (I_D_/I_G_) decreased with increases in RT, suggesting structural recovery. The four-point probe (4PP) showed that the TRGO at higher temperatures had a lower sheet resistance, i.e., it was highly conductive. The electrochemical performance showed that the TRGO-200 µ-SC had higher areal energy and lower areal power because more ions diffused into the internal surface, as suggested by the power law and Trasatti’s analysis. The TRGO-200 µ-SC had lower capacitance retention than the other µ-SCs due to the further reduction of the GO due to the repetitive GCD electrochemical measurements. The µ-SCs prepared at higher RT showed lower areal energy and higher areal power due to the narrow interplanar distance and a lack of OFGs preventing intercalation and promoting ion adsorption. Thus, lower and higher RT µ-SC are governed by diffusion- and surface-controlled behaviour, respectively.

## Figures and Tables

**Figure 1 nanomaterials-12-02211-f001:**
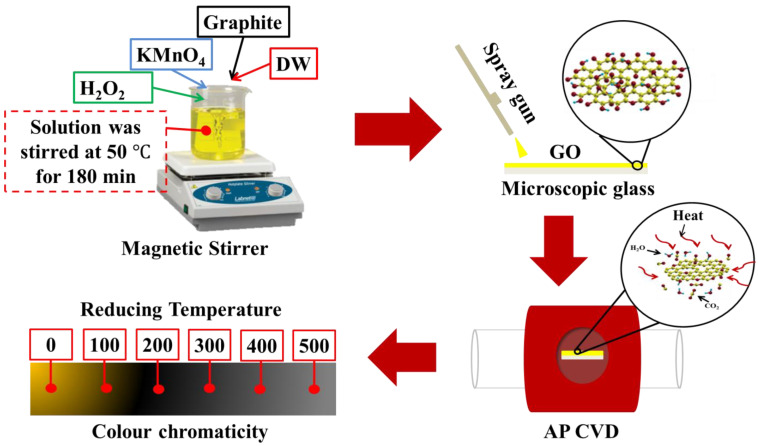
Schematic detailing TRGO thin-film preparation and the colour chromaticity showing the appearance of the prepared thin film.

**Figure 2 nanomaterials-12-02211-f002:**
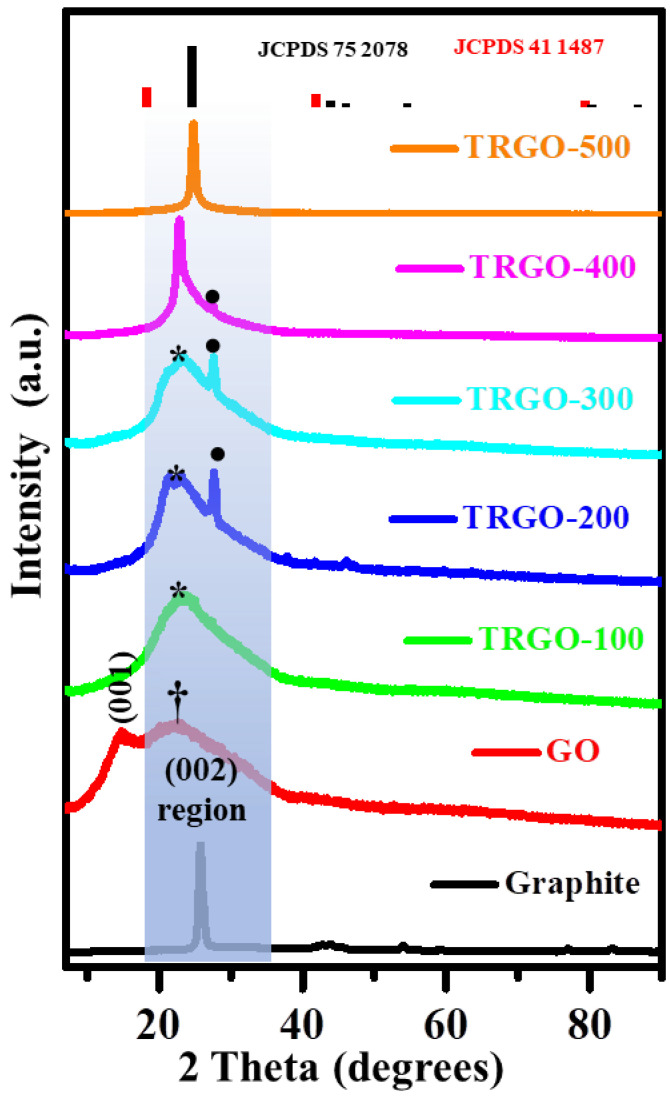
The XRD spectra for graphite, GO, and TRGO-100 to TRGO-500.

**Figure 3 nanomaterials-12-02211-f003:**
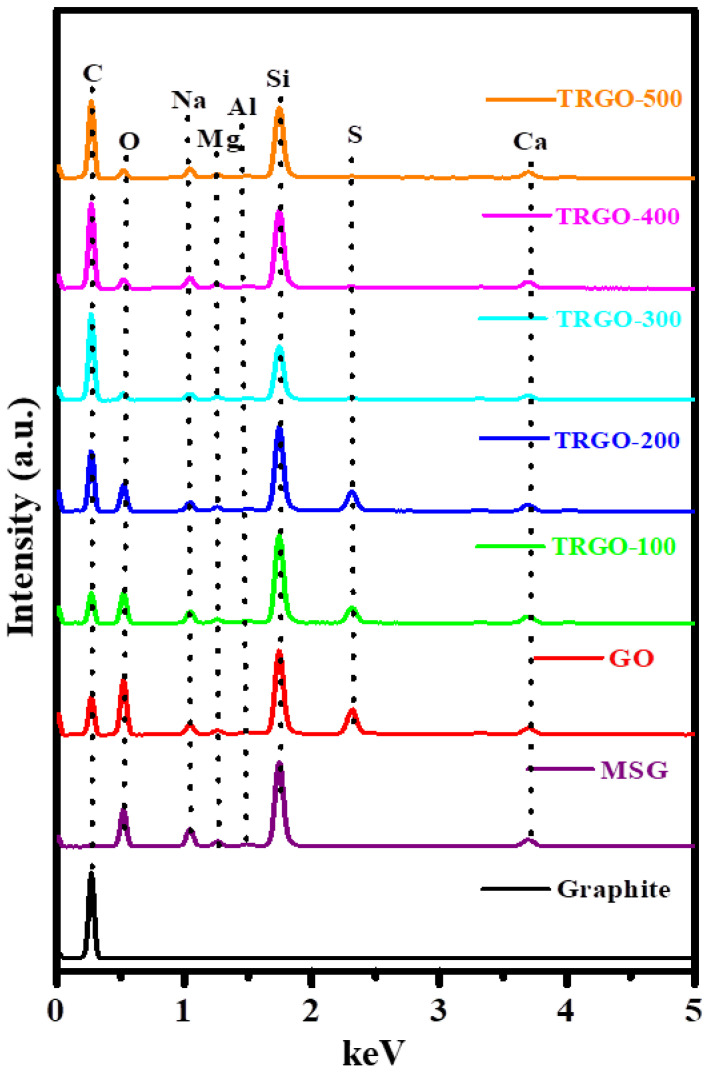
EDS spectrum of graphite, MSG, GO, and TRGO samples.

**Figure 4 nanomaterials-12-02211-f004:**
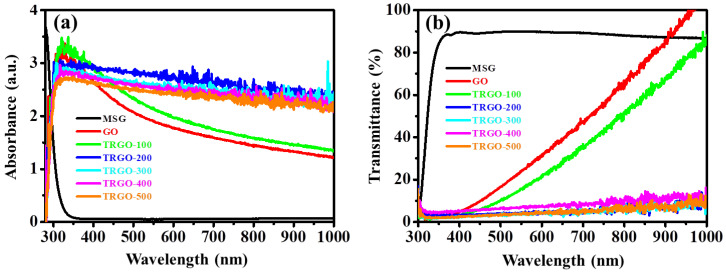
UV-Vis. (**a**) Absorbance and (**b**) transmittance of the GO, MSC, and TRGO-100 to TRGO–500.

**Figure 5 nanomaterials-12-02211-f005:**
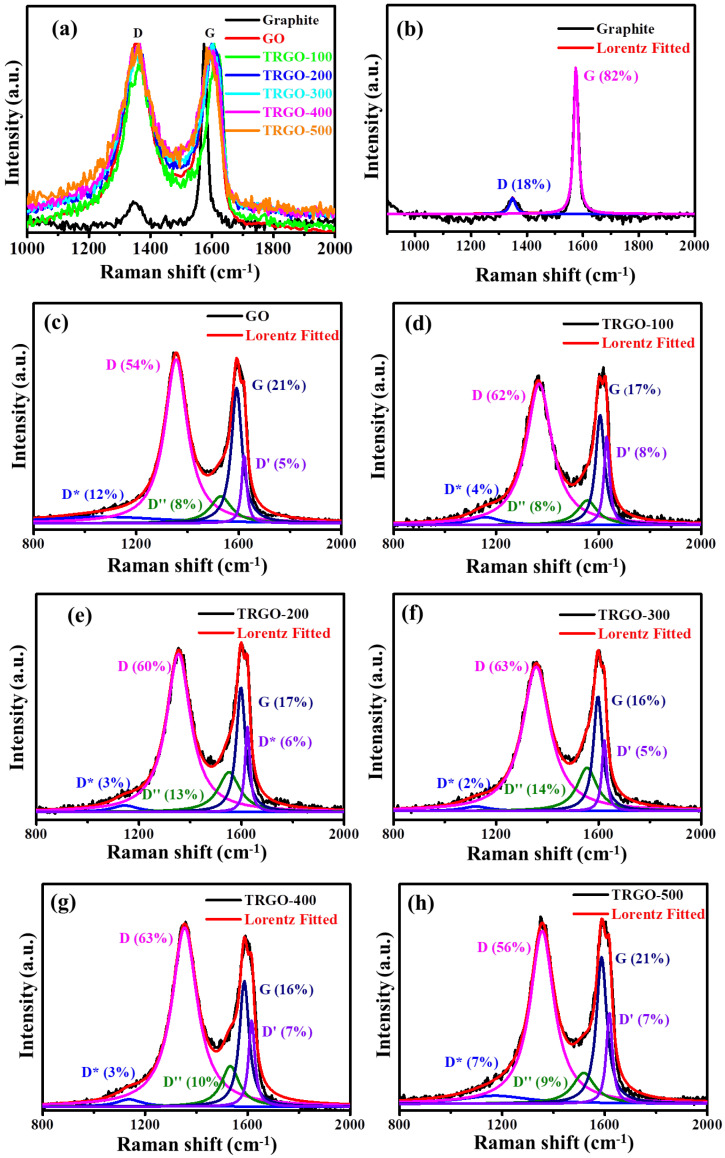
Raman spectra intensity of graphite, GO, and TRGO-100 to TRGO-500. One-phonon region of the Raman spectra of (**a**) graphite, (**b**) GO, and (**c**–**h**) TRGO-100–TRGO-500, plotted together with the best fit (R^2^ > 0.95 and ꭕ^2^ < 100) of the experiment to the sum of the five components (D*, D, D″, G, and D′).

**Figure 6 nanomaterials-12-02211-f006:**
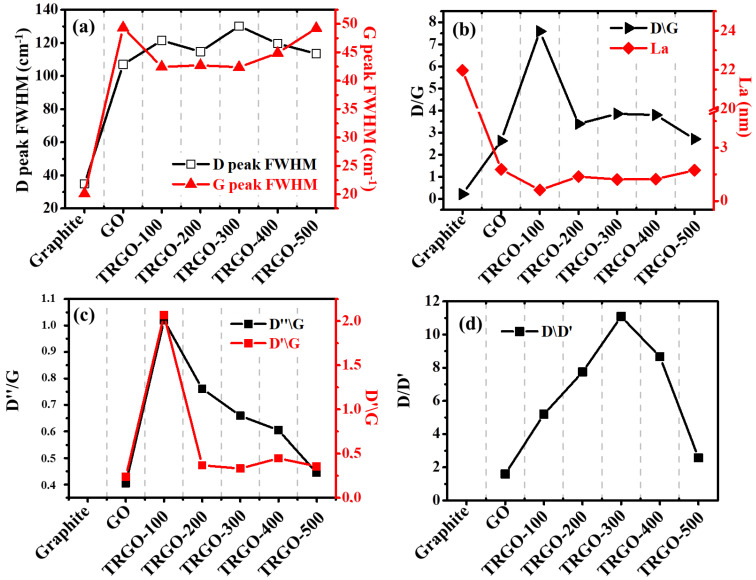
The (**a**) D- and G-peak FWHM, (**b**) D/G and La, (**c**) D″/G and (**d**) D/D′ as a function of graphite, GO, and TRGO-100 to TRGO-500.

**Figure 7 nanomaterials-12-02211-f007:**
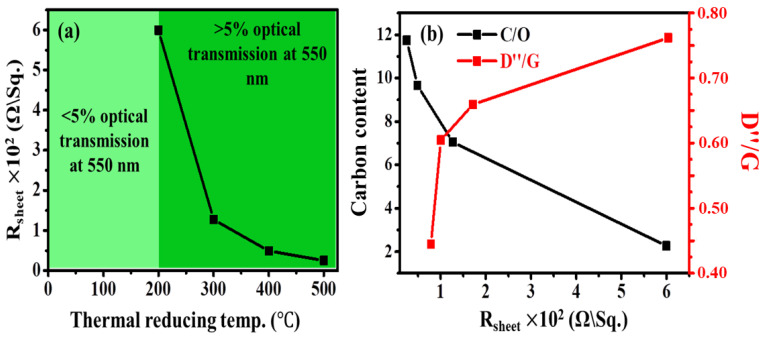
(**a**) Sheet resistance as a function of RT of the TRGO-200 to TRGO-500 and (**b**) carbon content and D″/G as function of sheet resistance.

**Figure 8 nanomaterials-12-02211-f008:**
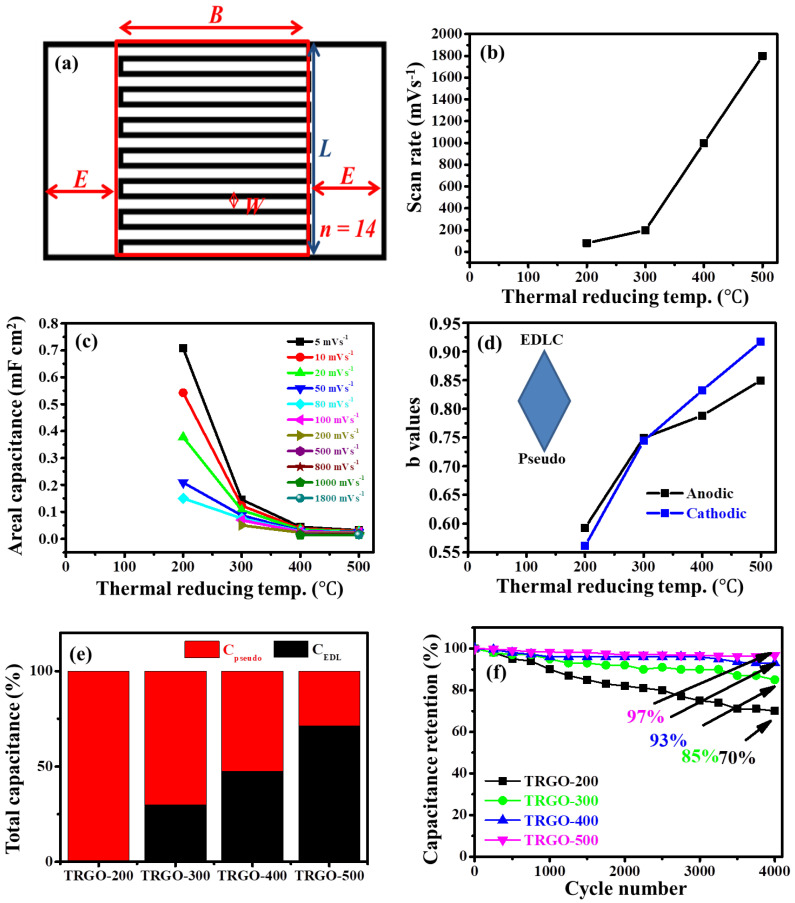
(**a**) µ-SC schematic, (**b**) scan rate, (**c**) areal capacitance, (**d**) b values versus RT, (**e**) Trasatti analysis capacitive contribution, and (**f**) capacitance retention as a function of cycle number of TRGO samples.

**Figure 9 nanomaterials-12-02211-f009:**
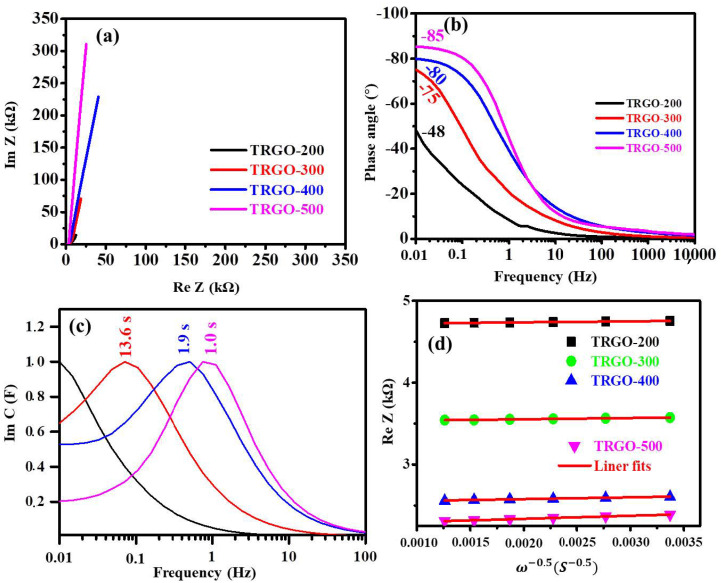
(**a**) EIS Nyquist plot, (**b**) phase angle versus frequency, (**c**) imaginary plot of capacitance as a function of frequency, and (**d**) real Z versus $\omega^{-0.5}$ at low frequencies for the TRGO samples, as labelled in the figures.

**Table 1 nanomaterials-12-02211-t001:** Comparison of electrochemical performance of carbon-based micro-devices.

Device Configuration	GO Reduction Method	Electrolyte	*V* (v)	CR (%)	Capacitance	*E_S_* @ *P_S_* @ *C_d_*	Ref.
TRGO-200 ^†^	Thermally treated at 200 °C in Ar for 10 min	PVA/H_3_PO_4_	0.8	95	0.7074 mF cm^−2^ @ 5 mVs^−1^	0.3329 mW h cm^−2^ @ 0.1093 mW cm^−2^ @ 8.3 mA cm^−2^	This work
TRGO-500 ^†^	Thermally treated at 500 °C in Ar for 10 min	PVA/H_3_PO_4_	0.8	70	0.0319 mF cm^−2^ @ 5 mVs^−1^	0.0380 mW h cm^−2^ @ 0.3663 mW cm^−2^ @ 8.3 mA cm^−2^	This work
rGO/MWCNT ^†^	788-nanometer-infrared-laser-treated at a power output of 5 mW	PVA/H_3_PO_4_	1	85.5	46.6 F cm^−3^	6.47 mW h cm^−3^ @ 10 mW cm^−3^ @ 20 mA cm^−3^	[50]
TRGO-coated fabric ^†^	Thermally treated at 160 °C in Ar for 2 h	PVA/H_3_PO_4_	1	93	70.4 F g^−1^ at 5 mV s^−1^	5.8 W h kg^−1^ @ 27.7 k W kg^−1^ @ 0.1 mA cm^−2^	[51]
pErGO@Cuf	Electrochemically reduced at an applied potential of −1.2 V for 20 CV cycles.	PVA/H_3_PO_4_	1	94.5	283.5 mF cm^−2^ at 0.5 Ag^−1^	39.3 μW h cm^−2^ @17.6 mW cm^−2^	[47]
Hydrothermally reduced GO	Hydrothermal method at 150 °C for 10 h in water	PVA/H_3_PO_4_-Na_2_MoO_4_	1	~100	38.2 mF cm^−2^	5.3 μW h cm^−2^	[52]
rGO/PEDOT	GO was laser-reduced and PEDOT was vapour-phase-polymerised onto rGO	PVA/H_3_PO_4_	0.8	90.2	35.12 Fcm^−3^ at 80 mAcm^−3^	4.876 mW h cm^−3^ @ 40 mW cm^−3^	[53]
LSG/MnO	Laser -educed at 532 nm	PVA/H_3_PO_4_	0.8	96	55 mF cm^−2^ @ 0.1 mA cm^−2^	4.89 μW h cm^−2^ @ 0.72 mW cm^−2^ @ 0.1 mA cm^−2^	[45]

†–symmetric device; *V*–voltage, CR–capacitor retention, rGO/MWCNT–reduced graphene oxide/multi-wall carbon nanotube; pErGO@Cuf–porous electrochemically reduced graphene oxide on cupper foam, PEDOT-poly(3,4-ethylenedioxythiophene), and LSG—laser-scan graphene.

## Data Availability

Data is contained within the article or Appendix A.

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
