# Peer review of "Impact of Thermally Reducing Temperature on Graphene Oxide Thin Films and Microsupercapacitor Performance"

_nanomaterials, 2022, doi:10.3390/nano12132211_

Round 1

Reviewer 1 Report

I would ask the authors to resubmit the manuscript. The manuscript is full of format errors. Difficult to read. Therefore, the authors can resubmit the manuscript after correcting the format. 

Author Response

I would ask the authors to resubmit the manuscript. The manuscript is full of format errors. Difficult to read. Therefore, the authors can resubmit the manuscript after correcting the format.

Response:

We thank the reviewer for raising this comment; the authors have poof-read the manuscript to ensure that the format errors are eradicated making the manuscript easy to read.

Reviewer 2 Report

1. I did not see any novel work that has been already published in 

Maphiri, V.M.; Rutavi, G.; Sylla, N.F.; Adewinbi, S.A.; Fasakin, O.; Manyala, N. Novel Thermally Reduced Graphene Oxide Microsupercapacitor Fabricated via Mask—Free AxiDraw Direct Writing. Nanomaterials 202111, 1909. https://doi.org/10.3390/nano11081909

https://doi.org/10.3390/nano11081909

Did not see any correlation between application and characterization used to explain application

Author Response

Did not see any correlation between application and characterization used to explain application

Response:

We thank the reviewer for raising this comment; the authors have poof-read the manuscript to ensure that the manuscript is properly written. With respect to our previous work (briefly), what is novel about that work is the entire microsupercapacitors preparation method i.e thermally reducing graphene oxide directly on the microscopic glass substrate without using additives binder and conductivity enhancer. This led to a very short preparation method, cheap and simple. On the contrary, an alternative route would be to independently reduced the prepared graphene oxide mix it with some binder the spray on the substrate glass, which would demand more time.

The correlation between the application and characterization explains why the low temperature reduced microsupercapacitors (LTRM) has more energy and low power density as compared to high temperature reduced microsupercapacitors (HTRM). In summary, XRD showed that LTRM has a higher graphene sheet spacing compared to HTRM which allows for more ion intercalation (diffusion) within the graphene sheet leading to more ion adsorption i.e. higher energy density. Techniques like the four point probe (4PP) showed that HTRM is more conductively hence it has a higher power density. Whereas the EIS analysis showed that the LTRM has high ion diffusion as compared to HTRM as also suggested by the XRD.

The preparation method have been accurately described on our previous published work, additional reference [20][21][22] have been added.

[20]    M.N. Rantho, M.J. Madito, N. Manyala, Symmetric supercapacitor with supercapattery behavior based on carbonized iron cations adsorbed onto polyaniline, Electrochim. Acta. 262 (2018) 82–96. https://doi.org/10.1016/j.electacta.2018.01.001.

[21]    V. Augustyn, P. Simon, B. Dunn, Pseudocapacitive oxide materials for high-rate electrochemical energy storage, Energy Environ. Sci. 7 (2014) 1597–1614. https://doi.org/10.1039/c3ee44164d.

[22]    V. Augustyn, J. Come, M.A. Lowe, J.W. Kim, P.L. Taberna, S.H. Tolbert, H.D. Abruña, P. Simon, B. Dunn, High-rate electrochemical energy storage through Li + intercalation pseudocapacitance, Nat. Mater. 12 (2013) 518–522. https://doi.org/10.1038/nmat3601.

Reviewer 3 Report

The manuscript nanomaterials-1774322 mainly presents a study about the synthesis and standard characterization of particular thermally reduced graphene oxide thin films. A microsupercapacitor performance is claimed. Please see below a list of comments to the authors:

  1. Besides the collective references selected in the report are suggested to be split in individual form to better present the topic, the motivation of the work must be better presented in order to improve the impact of the main conclusions.
  2. The data of about Ra, Sq, etc. parameters from the AFM analysis should be added.
  3. Please compare the XRD results in the sample studied with JCPDS data.
  4. Error bar in experimental results is mandatory.
  5. Please justify the correction constant in the Ohm law described by Eq. (3) for the samples studied.
  6. How is controlled the homogeneity or inhomogeneity in thermal energy transfer during the preparation of the samples?
  7. The size represents a strong influence on thermal effects. You can see for instance https://doi.org/10.1016/j.csite.2021.101453. Please argue.
  8. The potential influence of the impurities of the samples in the results should be described. And please report the expected impurities.
  9. A panoramic micrograph of the film would be welcome to see the homogeneity of the thickness and defects in the films.
  10. The main findings must be confronted with updated publications in the topic. The authors are invited to see for instance: https://doi.org/10.1364/OE.27.007330 and discuss within the text potential applications of their observations related to microcapacitance.

Author Response

The respond is in the attached file

Round 2

Reviewer 2 Report

The present format is acceptable

Author Response

  1. English language and style are fine/minor spell check required
  2. Does the introduction provide sufficient background and include all relevant references? Can be improved
  3. Are all the cited references relevant to the research? Needs to be improved
  4. Are the conclusions supported by the results? Can be improved

Comments and suggestions for authors: The present format is acceptable

Response:

We thank the reviewer for this wonderful comment that helped to bring out the novelty of our work. The introduction, reference and conclusion have been improved. 

Reviewer 3 Report

I appreciate the effort of the authors to address the points raised in the review stage. However the manuscript still presents some fundamental points that in my opinion should be addressed. Please see below:

  1. The references should be presented with individual expressions in order to justify their selection. The collective presentation of the references was not properly split. Moreover, the motivation of the work was not better presented and the conclusion was not improved.
  2. The correction constant in the Ohm law described by Eq. (3) was not analyzed neither justified for these particular samples presented in this work.
  3. The homogeneity or inhomogeneity in thermal energy transfer during the preparation of the samples was not analyzed.
  4. The main findings were not confronted with updated publications in the topic as suggested in order to see the value of the results and to improve the presentation of this work.
